# Environmental pH signals the release of monosaccharides from cell wall in coral symbiotic alga

Yuu Ishii[1,2†], Hironori Ishii[1], Takeshi Kuroha[1‡], Ryusuke Yokoyama[1], Ryusaku Deguchi[2], Kazuhiko Nishitani[3], Jun Minagawa[4,5], Masakado Kawata[1], Shunichi Takahashi[6], Shinichiro Maruyama[1,7*§]

[1]Department of Ecological Developmental Adaptability Life Sciences, Graduate School of Life Sciences, Tohoku University, Sendai, Japan; [2]Department of Biology, Miyagi University of Education, Sendai, Japan; [3]Department of Biological Sciences, Faculty of Science, Kanagawa University, Yokohama, Japan; [4]Department of Basic Biology, School of Life Science, SOKENDAI (The Graduate University for Advanced Studies), Okazaki, Japan; [5]Division of Environmental Photobiology, National Institute for Basic Biology, Okazaki, Japan; [6]Tropical Biosphere Research Center, University of the Ryukyus, Okinawa, Japan; [7]Graduate School of Humanities and Sciences, Ochanomizu University, Tokyo, Japan

*For correspondence:
shinichiro.maruyama@k.u-tokyo.ac.jp

Present address: †Graduate School of Agriculture, Kyoto University, Sakyo-ku, Japan; ‡Division of Crop Genome Editing, Institute of Agrobiological Sciences, National Agriculture and Food Research Organization (NARO), Tsukuba, Japan; §Department of Integrated Biosciences, Graduate School of Frontier Sciences, The University of Tokyo, Kashiwa, Japan

Competing interest: The authors declare that no competing interests exist.

**Abstract** Reef-building corals thrive in oligotrophic environments due to their possession of endosymbiotic algae. Confined to the low pH interior of the symbiosome within the cell, the algal symbiont provides the coral host with photosynthetically fixed carbon. However, it remains unknown how carbon is released from the algal symbiont for uptake by the host. Here we show, using cultured symbiotic dinoflagellate, *Breviolum* sp., that decreases in pH directly accelerates the release of monosaccharides, that is, glucose and galactose, into the ambient environment. Under low pH conditions, the cell surface structures were deformed and genes related to cellulase were significantly upregulated in *Breviolum*. Importantly, the release of monosaccharides was suppressed by the cellulase inhibitor, glucopyranoside, linking the release of carbon to degradation of the agal cell wall. Our results suggest that the low pH signals the cellulase-mediated release of monosaccharides from the algal cell wall as an environmental response in coral reef ecosystems.

## Editor's evaluation

The manuscript makes a fundamental contribution to our understanding of sugar release by symbiotic dinoflagellates and is of broad interest to the fields of ecology, marine biology, and cell biology. The experiments, which combine algal culture with targeted metabolomics, transcriptomics, and the application of inhibitors, provide convincing evidence for an acidic environment mimicking conditions reported for the intracellular organelle that hosts the symbiotic algae, leading to upregulation of algal cellulases, which in turn degrade the algal cell wall and thereby releasing glucose and galactose that can be used as a source of food by the coral host. This is a new idea and could significantly contribute to our understanding of photosymbiosis.

## Introduction

Coral reef ecosystems are sustained by symbiosis between stony corals and marine dinoflagellates from the family Symbiodiniaceae, which are found in nature as free-living mixotrophs (*Decelle et al.,*

**eLife digest** Coral reefs are known as 'treasure troves of biodiversity' because of the enormous variety of different fish, crustaceans and other marine life they support. Colonies of marine animals, known as corals, which are anchored to rocks on the sea bed, form the main structures of a coral reef. Many corals rely on partnerships with microscopic algae known as dinoflagellates for most of their energy needs. The dinoflagellates use sunlight to make sugars and other carbohydrates and they give some of these to the coral. In exchange, the coral provides a home for the dinoflagellates inside its body.

The algae live inside special compartments within coral cells known as symbiosomes. These compartments have a lower pH (that is, they are more acidic) than the rest of the coral cell. Previous studies have shown that the algae release sugars into the symbiosome but it remains unclear what triggers this release and whether it only occurs when the algae are in a partnership.

Ishii et al. studied a type of dinoflagellate known as *Breviolum sp.* that had been grown in sea water-like liquid in a laboratory. The experiments found that the alga released two sugar molecules known as glucose and galactose into its surroundings even in the absence of a host coral.

Increasing the acidity of the liquid caused the alga to release more sugars and resulted in changes to some of the structures on the surface of its cells. The alga also produced an enzyme, called cellulase, to degrade the wall that normally surrounds the cell of an alga. Treating the alga with a drug that inhibits the activity of cellulase also suppressed the release of sugars from the cells.

These findings suggest that when dinoflagellates enter acidic environments, like the guts of marine animals or symbiosomes inside coral cells, the decrease in pH can activate the algal cellulase enzyme, which in turn triggers the release of sugars for the coral. This research will provide a new viewpoint to those interested in how partnerships between animals and algae are sustained in marine environments. It also highlights the importance of the alga cell wall in establishing partnerships with corals. Further work will seek to clarify the precise biological mechanisms involved.

*2018*; *Jeong et al., 2012*), as well as are primary producers in symbiotic relationships with various partners, including multicellular (e.g. Cnidaria, Mollusca, Porifera) and unicellular organisms (Foraminifera, ciliates; *LaJeunesse et al., 2018*). In oligotrophic oceans, transfer of atmospheric carbon photosynthetically fixed by the symbiotic algae to their hosts is a fundamental flux to sustain the growth and productivity of coral reef ecosystems.

Although it is generally accepted that Symbiodiniaceae algae provide photosynthates to their symbiotic partners, the molecular details are largely unknown (*Falkowski et al., 1984*; *Ishii et al., 2019*; *Ishikura et al., 1999*; *Muscatine, 1990*; *Rahav et al., 1989*; *Stat et al., 2008*; *Whitehead and Douglas, 2003*). Members of this family reside in the extracellular 'symbiotic tube' systems of giant clams or in an intracellular organelle called the 'symbiosome' within cnidarian host cells. These are thought to be special low pH environments that are acidified by V-type $H^+$-ATPase proton pumps (*Armstrong et al., 2018*; *Barott et al., 2015*; *Davy et al., 2012*). While low pH environments are stressors to algae in general, they can be beneficial when $CO_2$ uptake is encouraged by the hosts' carbon-concentrating functions, enhancing photosynthesis (*Armstrong et al., 2018*; *Barott et al., 2015*). A previous study has demonstrated a photosynthesis-dependent glucose transfer from Symbiodiniaceae to sea anemone hosts (*Burriesci et al., 2012*), and some sugar transporters are proposed to be involved in glucose transfer (*Lehnert et al., 2014*; *Sproles et al., 2018*). Other studies suggest that the amount of transfer is regulated by the C-N balance (*Rädecker et al., 2021*; *Xiang et al., 2020*). Nevertheless, the mechanism of algal glucose secretion is not yet characterized.

As walled organisms, microalgae respond to the environments in a variety of ways through their cell walls. Although dinoflagellates including Symbiodiniaceae have cellulose-containing cell walls that are structurally distinct from those of land plants, molecular organization of the cell walls is poorly understood. Previous studies have shown that enzymes involved in the degradation and synthesis of cellulose (e.g. Cellulase, cellulose synthase) are critical in the regulations of the cell cycle and cell morphology, suggesting that the cell wall is a dynamic environmental interface (*Chan et al., 2019*; *Kwok and Wong, 2010*). In this study, we focus on the responses to low pH and the cell wall

organization of the coral symbiont alga *Breviolum* sp., which provides insights into what roles the simple environmental responses can play in broader contexts, including symbiosis.

## Results

To investigate the physiological effects of low pH, a characteristic environmental factor in symbioses, on algal intrinsic properties, a Symbiodiniaceae alga *Breviolum* sp. SSB01 (hereafter, *Breviolum*) was grown in a host-independent manner and cell proliferation and photosynthetic activities were measured (*Figure 1—figure supplement 1*). By comparing the growth rate of *Breviolum* in normal culture medium (pH 7.8) and acidic medium (pH 5.5, hereafter called 'low pH'), we showed that the low pH medium considerably suppressed algal growth (*Figure 1A*) and the cells in low pH media were more spread out and less clustered than the cells in normal media (*Figure 1B and C*). In addition, culturing at low pH for 1 day resulted in significant declines in photosynthesis activity (*Figure 1D*).

Contrary to our expectation, the amount of glucose secreted into the culture medium was higher at low pH (*Figure 1E*) and the secreted galactose similarly showed an increasing trend (*Figure 1F*). These trends suggest that *Breviolum* is capable of secreting monosaccharides autonomously without host signals, and that low pH enhanced the secretion. On the addition of the photosynthesis inhibitor 3-(3,4-dichlorophenyl)–1,1-dimethylurea (DCMU), the concentrations of glucose and galactose in the medium increased (*Figure 1E and F*, *Figure 1—figure supplement 2*), suggesting the presence of a pathway uncharacterized in previous studies, where the transport of newly fixed glucose, not glycerol, to the host sea anemone was blocked by DCMU addition (*Burriesci et al., 2012*).

To investigate the response of *Breviolum* to acidic environments at the morphological level, cells cultured in different media were examined by microscopy (*Figure 1—figure supplement 1*). Scanning electron microscope (SEM) observations revealed that many of the *Breviolum* cells cultured at low pH exhibited wrinkled structures on their cell surfaces (*Figure 2A and B*). Furthermore, transmission electron microscopy (TEM) revealed that the cell surface structures of the low pH media group were more 'exfoliated' (*Figure 2C and D*). These suggest that low pH affects the structures and properties of a cellulosic cell wall found in coccoid Symbiodiniaceae cells (*Colley and Trench, 1983*; *Markell et al., 1992*).

To identify the mechanism involved in the monosaccharide secretion of *Breviolum*, we compared gene expression changes between the 'control vs normal' and 'control vs low pH' comparisons (*Figure 1—figure supplement 1*), and identified 3 and 4527 differentially expressed genes (DEGs), respectively (*Figure 3A*, *Figure 3—source data 1*). The gene ontology (GO) term enrichment and KEGG pathway analysis of these two gene sets resulted in the detection of 0 (control vs normal) and 16 (control vs low pH) terms (*Figure 3—source data 2*), which included categories related to carbon metabolism (*Figure 3B*, *Figure 3—figure supplement 1*). The CAZy database (*Lombard et al., 2014*) analysis showed that 12 DEGs (28 isoforms) were annotated with Carbohydrate-Active enZymes (CAZymes) activity (*Figure 3—source data 3*). One of the the gene models, TRINITY_DN40554_c2_g2, was shown to encode Glycoside Hydrolase Family 7 (GH7) endo-β–1,4-glucanase (exocrine cellulolytic enzyme) harbouring a signal peptide and a sequence motif called Carbohydrate-Binding Module Family 1 (CBM1) (*Figure 3C*) with high similarity to dinoflagellate cellulases (*Kwok and Wong, 2010*; *Figure 3—figure supplement 2*). Among four isoforms of this cellulase gene annotated as GH7, one lacked the N-terminal region including a signal peptide and CBM1 motif (labelled as 'GH7 +CBM1' in *Figure 3C*), but the rest of the sequences were highly conserved at the amino acid level and only distinguished by small variations. Notably, this cellulase gene was detected as a DEG in the comparison between free-living and symbiotic algae using the published dataset (*Figure 3—source data 1*).

To confirm the effect of cellulase on monosaccharide secretion, we examined whether secretions were inhibited by the cellulase inhibitor Para-nitrophenyl 1-thio-beta-d-glucopyranosid (PSG) (*Yoshida, 1995*). Prior to examining this, we confirmed the inhibitory effect of PSG on cellulase activity in *Breviolum* cells in vitro. Although the cellulase activity in the cell supernatant was too low to be detected, PSG inhibited the cellulase activity in *Breviolum* cell homogenate in a concentration-dependent manner (*Figure 4—figure supplement 1*). Then, we examined the effect of PSG on the amount of glucose and galactose secreted in vivo using the cell cultures under low pH (*Figure 1—figure supplement 1*). PSG inhibited the secretion of both glucose and galactose in a dose-dependent manner (*Figure 4*), suggesting that degradation of the cell wall containing glucose and galactose by cellulase is involved in the secretion of monosaccharides from *Breviolum* cells.

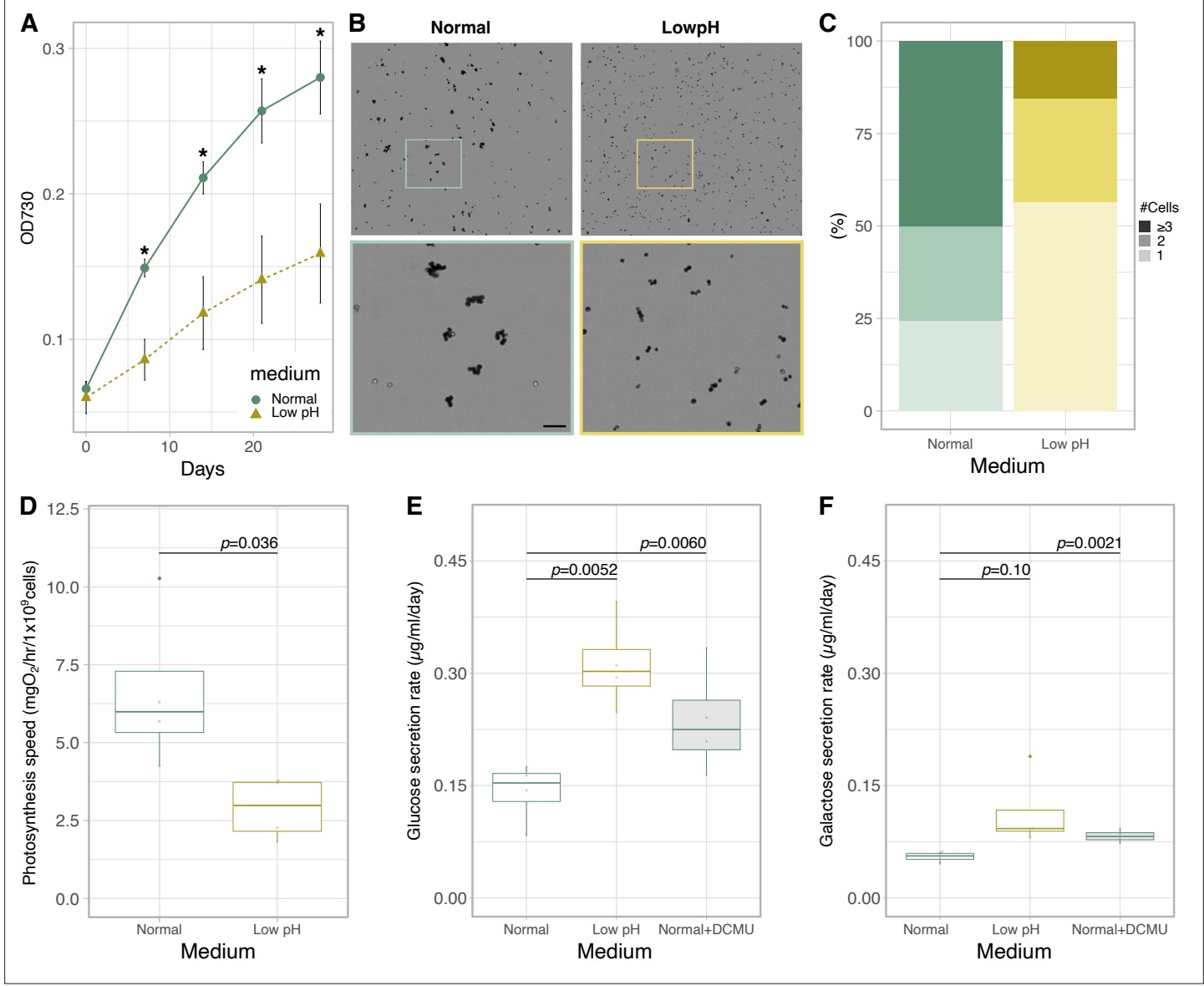

**Figure 1.** Physiological characterization and monosaccharide secretion of cultured *Breviolum*. (**A**) Growth rate (n = 6 per treatment, t-test). Asterisks indicate statistically significant differences (t-test, *p* < 0.005). (**B**) Bright field images of the cells under different conditions. The lower panels show high-magnification views of boxed areas in the upper panels. Scale bar = 50 μm. (**C**) Quantification of the number of cells forming clusters (Fisher's exact test for "1 or 2" vs "3 or more", *p* = 1.727 × 10$^{-7}$). (**D**) Photosynthesis activity (n = 4 per treatment, t-test) (**E, F**) Quantification of glucose (**E**) and galactose (**F**) secreted in normal, low pH and normal+DCMU media during incubation for 1 day using ion chromatography (n = 4 per treatment, t-test).

The online version of this article includes the following source data and figure supplement(s) for figure 1:

**Figure supplement 1.** Experimental design details.

**Figure supplement 2.** Effect of photosynthesis inhibitor on glucose and galactose secretion.

**Source data 1.** Raw data of growth rates.

**Source data 2.** Raw data of photosynthesis activities.

**Source data 3.** Raw data of glucose and galactose concentrations.

# Discussion

The transfer of photosynthetically fixed carbon from symbiotic algae to host cnidarians, including corals, is a cornerstone of their mutual symbiotic relationship (*Muscatine, 1990*). Unlike the current accepted model of monosaccharide secretion that assumes photosynthetically fixed carbon is directly

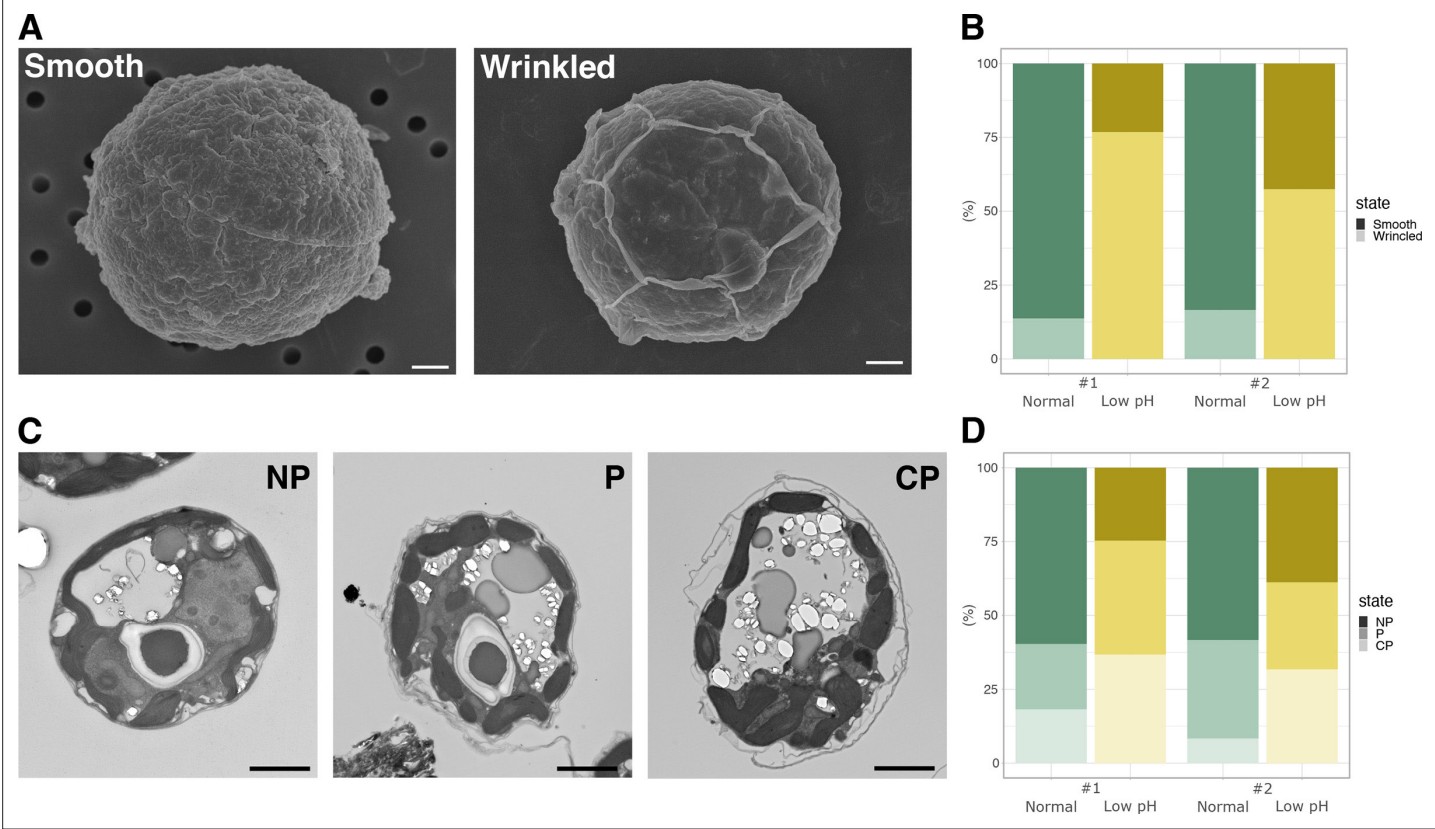

**Figure 2.** Cell structures under different pH condition. (**A**) SEM images of the representative cells. Scale bar = 1 µm. (**B**) Quantification of the cell surface structures of the SEM images (Fisher's exact test, #1; $p < 2.2 \times 10^{-16}$, #2; $p < 2.2 \times 10^{-16}$). (**C**) TEM images of the representative cells. NP, 'non-peeled' where the outer struc- ture of the cell wall is not shed from the cell surface; P, 'peeled' at some parts of the cell surface; CP, 'completely peeled'. Scale bar = 2 µm. (**D**) Quantification of the cell surface structures of the TEM images (Fisher's exact test for "P or CP" vs "NP", #1; $p = 4.621 \times 10^{-11}$, #2; $p = 3.525 \times 10^{-4}$).

The online version of this article includes the following source data and figure supplement(s) for figure 2:

**Figure supplement 1.** SEM images of cell surface structures.

**Figure supplement 2.** TEM images of cell surface structures.

**Source data 1.** Raw data of bright field cell counts.

**Source data 2.** Raw data of SEM cell counts.

**Source data 3.** Raw data of TEM cell counts.

exported from the algal symbiont via unidentified glucose transporter(s) (*Lehnert et al., 2014*; *Sproles et al., 2018*), our results suggest that stored carbon can be released from the algal cell wall as an environmental response. In the present study, we showed that a decrease in ambient pH, consistent with the interior pH of the symbiosome, accelerates the monosaccharide secretion from *Breviolum* (*Figure 1*). Importantly, this low pH-associated secretion was suppressed by inhibition of cellulase (*Figure 4*), suggesting that algal symbionts release monosaccharides into the symbiosome within host cnidarian cells by cell wall degradation. Previous studies showed that cell wall degrada- tion/rearrangement by cellulase is required for cell cycle progression (*Kwok and Wong, 2010*) and cellulose synthesis is involved in morphogenesis (*Chan et al., 2019*) in dinoflagellates. Indeed, wrinkle and exfoliation of the algal cell wall was observed under low pH conditions using SEM and TEM, respectively, suggesting that the cell walls are morphologically and qualitatively modified under low pH in *Breviolum* (*Figure 2*). We need to note that our results do not deny the current accepted model, but rather suggest a multi-pathway hypothesis supported by the following observations: (i) the secre- tion of monosaccharides was not completely inhibited by cellulase inhibition, (ii) this new pathway occurred in a day, compared to previous reports, where exported glucose was detectable in the host after only 30–60 mins (*Burriesci et al., 2012*), (iii) in contrast to previous studies (*Burriesci et al.,*

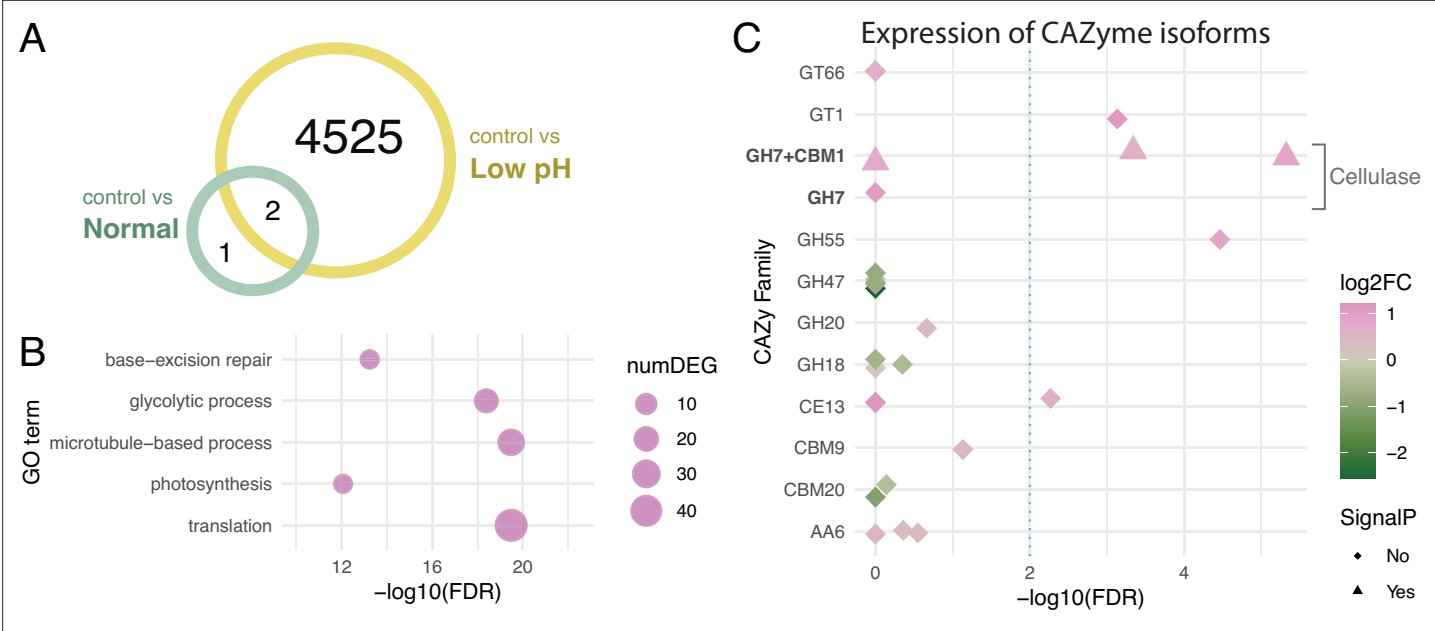

**Figure 3.** Differentially expressed genes under different pH conditions. (**A**) Venn diagram showing the numbers of DEGs under different conditions. (**B**) GO term enrichment analysis. Circles indicate the statistical significance (FDR) of the enriched GO terms, with the numbers of DEGs (numDEG) associated with each GO term. (**C**) Isoform-level expression analysis of genes encoding Carbohydrate-Active enZymes (CAZymes). Symbols indicate isoforms associated with the DEGs, with the presence (triangle) or absence (rhombus) of signal peptide predicted in the amino acid sequence. Symbol colors represent the log2 fold-changes (log2FC) of the expression levels of each isoform (low pH/control). Dashed line indicates a threshold for differential expression (FDR = 0.01).

The online version of this article includes the following source data and figure supplement(s) for figure 3:

**Figure supplement 1.** Mapping of DEGs between the low pH and control groups on KEGG pathways.

**Figure supplement 2.** Phylogenetic tree of the cellulase proteins.

**Source data 1.** Expression levels of the annotated low pH DEGs.

**Source data 2.** GO enrichment analysis results.

**Source data 3.** CAZome analysis results.

**Source data 4.** Multiple alignment of the cellulase proteins.

**Source data 5.** Newick format file for phylogenetic tree of the cellulase proteins.

*2012*), DCMU did not suppress, but rather increased the secretion of monosaccharides over a longer time span (*Figure 1E and F*). Importantly, low pH upregulated the expression of genes associated with not only cellulase (*Figure 3*) but glycolysis probably fuelled by degradation of storage compound like starch (*Figure 3—figure supplement 1*). This suggests the photosynthesis-limiting conditions trigger an environmental response in the algae to compensate for retarded cell cycle progression by upregulating multiple genes including the one encoding cellulase, accompanying cell wall degradation and monosaccharide secretion (or "leakage").

Like Symbiodiniaceae, some freshwater green algae are known to be symbiotic with a range of hosts. A number of *Chlorella* strains, with and without symbiotic ability, autonomously secrete monosaccharides under low pH conditions via unknown mechanisms (*Kessler et al., 1991*; *Mews and Smith, 1982*). The monosaccharides include maltose and, to a lesser extent, glucose (*Arriola et al., 2018*). Some dinoflagellates are also known to secrete viscous substances, including monosaccharides, as an environmental response, likely for cell aggregation and biofilm formation (*Kwok et al., 2023*; *Mandal et al., 2011*). In this study, we show that *Breviolum* secrets galactose as well as glucose (*Figure 1*). Although the mechanism of action of galactose secretion is unknown, less substantial increase of galactose secretion under low pH (*Figure 1*) and the significant inhibitory effect of PSG (*Figure 4*) suggest that galactose secretion may be regulated by uncharacterized PSG-sensitive enzymes. Under low pH, multiple genes encoding CAZymes that break down glycosidic bonds (e.g.

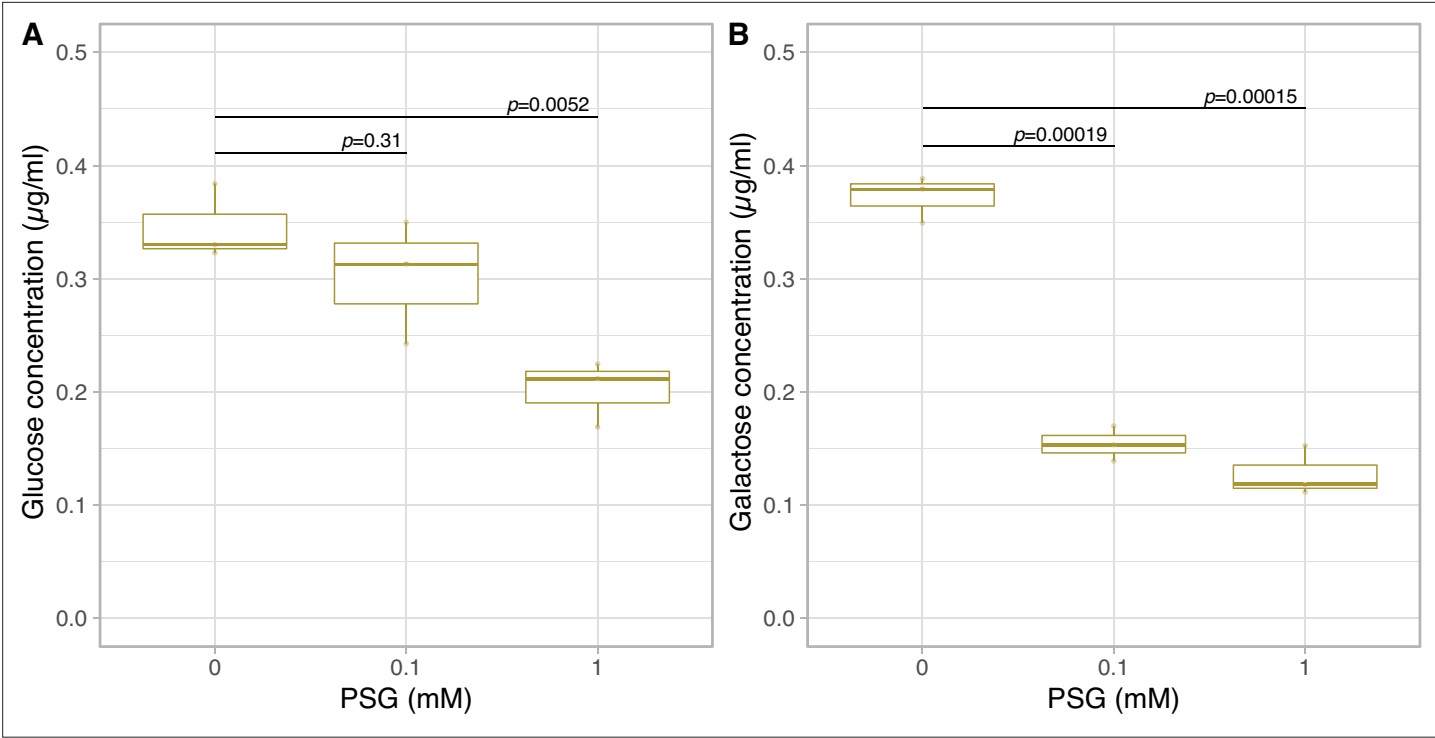

**Figure 4.** Glucose and galactose secretion in cellulase inhibitor treatment. The quantification of glucose (**A**) and galactose (**B**) in the medium on 1 day incubation with PSG using LC-MS/MS (n = 3 per treatment, t-test).

The online version of this article includes the following source data and figure supplement(s) for figure 4:

**Figure supplement 1.** Effect of cellulase inhibitor on the *Breviolum* cellulase activity in vitro.

**Source data 1.** Raw data of glucose and galactose concentrations with PSG.

**Source data 2.** Raw data of cellulase activity in vitro.

chitinase, hexosaminidase, mannosidase) were upregulated (*Figure 3*). The cell wall components of Symbiodiniaceae are unknown, but complex galactose-containing glycans that constitute the cell wall may be targets of these CAZymes. Overall, in microalgae, although the repertoire of molecular species secreted and the ecological consequences may vary, secretion of carbon as a form of saccharide appears to be a fairly conserved environmental response relevant to cell physiology and proliferation. Therefore, acidic symbiosomes may be of evolutionary advantage for cnidarian hosts to promote environmental responses of algal symbionts, which enables monosaccharides to be efficiently secreted within the organelle.

Generally, within ecosystems energy is transferred from photosynthetic primary producer to consumer by predation. Uniquely, in coral reef ecosystems energy is mainly transported from algae to corals after establishing a symbiotic relationship (*Davy et al., 2012*). Thus, understanding its mechanism has wider implications to understanding how energy is shared over the entire coral reef ecosystem. The multi-pathway hypothesis we propose here entails the direct transfer of photosynthates via glucose transporter(s) on their cell membrane (*Lehnert et al., 2014*; *Sproles et al., 2018*) as well as monosaccharide secretion following cell wall degradation. It remains to be determined how much each pathway contributes to the energy supply of host. However, since one pathway uses de novo photosynthates and the other uses stored photosynthates, combined they might allow for a stable supply of energy to the host, for example, over the entire light/dark day cycle, and under photosynthesis-limiting conditions like environmental stress or cloudy days where the cellulose-related pathway could be of substantial importance. Although genetic transformation and cell wall characterization of Symbiodiniaceae is still developing, the cellulase gene knock-out may bring a clue to test this (*Chen et al., 2019*; *Gornik et al., 2022*). Overall, our study provides a new insight into how carbon is provided by symbiotic algae to the coral reef ecosystem.

## Materials and methods

### Strains and culture conditions

We obtained the *Breviolum* (formerly *Symbiodinium* clade B) strain SSB01, an axenic uni-algal strain closely related to the genome-sequenced strain *B. minutum* Mf1.05b (clade B), as a generous gift from Profs. John R. Pringle and Arthur R. Grossman (*Shoguchi et al., 2013*; *Xiang et al., 2013*). The *Breviolum* was maintained according to previous study (*Ishii et al., 2018*). Stock cultures were incubated at 25°C in medium containing 33.5 g/L of Marine Broth (MB) (Difco Laboratories, New Jersey, USA), 250 mg/L of Daigo's IMK Medium (Nihon Pharmaceutical, Japan), and PSN (Gibco, Thermo Fisher Scientific, Massachusetts, USA), with final concentrations of penicillin, streptomycin, and neomycin at 0.01, 0.01, and 0.02 mg/mL, respectively. Light was provided at an irradiance of approximately 100 µmol photons/m$^2$s in a 12 hr light:12 h dark cycle. In experiments, IMK medium containing 33.5 g/L of sea salt (Sigma-Aldrich, Merck Millipore, Germany), 250 mg/L of Daigo's IMK Medium, and PSN with final concentrations of penicillin, streptomycin, and neomycin at 0.01, 0.01, and 0.02 mg/mL, respectively, was used as normal medium (pH 7.8). For the low pH experiments, the pH was adjusted to 5.5 using HCl to make low pH medium (pH 5.5). Prior to measurements, *Breviolum* was pre-incubated in normal medium for one week unless otherwise specified (*Figure 1—figure supplement 1*). *Breviolum* was pre-incubated in normal medium for one week unless otherwise specified (*Figure 1—figure supplement 1*).

### Growth rate, cell clumping and photosynthesis activity assay

*Breviolum* cultures were inoculated to fresh normal or low pH media for four weeks to measure growth rate (n=6 biological replicates). Growth rate comparisons between the normal and low pH media conditions were conducted using 100 µL of media (625 cells/µL) in a 96-well plate. Cell growth was monitored by measuring the optical density at 730 nm (OD730) of the liquid cultures using a Multiskan GO microplate spectrophotometer (Thermo Fisher Scientific, Massachusetts, USA) for once par week.

To compare the cell clumping conditions, *Breviolum* cultures were inoculated to fresh normal or low pH media for 3 weeks (12 hr light:12 hr dark). *Breviolum* cells were cultured starting at densities of $1.6 \times 10^7$ cells/20 mL per T25 culture flask. Cell photos were taken using a TC20 automated cell counter (Bio-Rad Laboratories, Hercules, CA), and the numbers of cells adjacent to and isolated from other cells were randomly counted (853 and 664 cells pooled form n=3 biological replicates were scored in the normal and low pH conditions, respectively).

To measure photosynthesis activities, *Breviolum* cultures were inoculated to fresh normal or low pH media for 1 day (n=4 biological replicates). Photosynthesis and respiration rates were measured with a Clark-type oxygen electrode (Hansatech Instruments, Norfolk, UK) in a closed cuvette under light at 1,000 µmol/m$^2$s photons at 25°C. The cultures were preincubated in the dark for 10 min and then exposed to saturating light for 20 min. Photosynthesis activities were determined using cultures at densities of $1 \times 10^6$ cells/mL in fresh normal and low pH media, on days 0 and 1 after changing the medium. Respiration rates were calculated using the dark-phase oxygen consumption rates and photosynthesis rates were calculated by subtracting the respiration rates from the light-phase oxygen evolution rates. Mean estimates with standard errors were calculated from single measurements of four different cultures per medium condition.

### Ion chromatography

*Breviolum* cultures were inoculated to fresh normal or low pH media for 1 day to measure the concentrations of monosaccharides. *Breviolum* cultures were incubated at densities of $8 \times 10^6$ cells/20 mL in T25 culture flasks with a filter cap (TrueLine Cell Culture Flasks, TR6000) under a light-dark cycle. DCMU (Tokyo Chemical Industry, Japan) was dissolved in ethanol at the concentration of 20 mM and was added to cultures to a final concentration of 20 µM followed by 1 day incubation, while the control samples contained the same amount of ethanol. The cells cultured with and without DCMU for 1 day were removed by centrifugation at 2000×*g* for 5 min at room temperature. Samples (n=4 biological replicates) of the supernatant from the control (0 day), normal (1 day) and low pH (1 day) cultures were filtered using a 0.22 µm PVDF filter (Merck Millipore, Germany). These samples were loaded onto an OnGuard column (Dionex OnGuard II Ba/Ag/H 2.5 cc Cartridge) (Thermo Fisher Scientific) to remove the sulphate and halogen, according to the manufacturer's instructions. The samples were quantified using high-performance anion-exchange chromatography with pulsed amperometric

detection (HPAEC-PAD) using a Dionex ICS-5000 system equipped with a CarboPac PA1 column (Dionex) (*Shinohara et al., 2017*). The column was operated at a flow rate of 1.1 mL/min with the following phases: (1) a linear gradient of 0–100 mM NaOH from 0 to 31 min, (2) a linear gradient of 0–150 mM sodium acetate containing 100 mM NaOH from 31 to 34 min, and (3) an isocratic 150 mM sodium acetate/100 mM NaOH from 34 to 41 min. Myo-inositol (2 μg/mL) was added to each sample as an internal standard for quantification.

The concentrations of monosaccharides were calculated by comparing the peak ratios between the targets of interest and standards. The secretion rates were calculated by subtracting the concentrations on 0 day from those of 1 day.

## Electron microscopy

*Breviolum* cultures were inoculated to fresh normal or low pH media for 1 day to examine morphological change. *Breviolum* cultures were incubated at densities of $8 \times 10^6$ cells/20 mL in T25 culture flasks with a filter cap (TrueLine Cell Culture Flasks, TR6000) under a light-dark cycle. For SEM observation, cells were fixed in 2% glutaraldehyde and 2% osmium (VIII) oxide, dehydrated with ethanol, and dried using the critical point drying technique. The samples were coated with osmium plasma and observed under a JSM-7500F microscope at 5 kV (Hanaichi UltraStructure Research Institute, Japan). The surface patterns of the cells were manually scored and classified as 'Smooth' or 'Wrinkled' (73 and 129 cells pooled form n=4–5 technical replicates in each of biological replicates (n=2) were blindly scored under the normal and low pH conditions, respectively). For TEM observation, cells were fixed in 2% glutaraldehyde and 2% osmium (VIII) oxide, dehydrated with ethanol and embedded in EPON812 polymerized with epoxy resin. Sections 80–90 nm thick were cut, coated with evaporated carbon for stabilisation, and stained with uranyl acetate and lead citrate. The sections were then imaged at 100 kV using a HITACHI H-7600 transmission electron microscope (Hanaichi UltraStructure Research Institute, Japan). The cells were then categorized as NP (non-peeled), P (peeled) or CP (completely-peeled) (cells from two pairs of biological replicates under normal and low pH conditions, pooled form n=10 technical replicates for each, were blindly scored).

## RNA extraction and sequencing

After pre-incubated in normal media for one week, *Breviolum* cultures were inoculated to fresh normal or low pH IMK media for 1 day to examine the transcriptional change. *Breviolum* cultures were incubated at densities of $8 \times 10^6$ cells/20 mL in T25 culture flasks with a filter cap (TrueLine Cell Culture Flasks, TR6000) under a light-dark cycle. The cultured cells were collected by centrifugation at $2000 \times g$ for 5 min at room temperature. Four samples (n=4 biological replicates) from each of the control (day 0), normal (day 1), or low pH (day 1) cultures were added to 500 μL of TRIZOL reagent (Thermo Fisher Scientific, Massachusetts, USA) and stored at –80°C. The samples were ground with two sizes of glass beads (20 μL volume each of '≤106 μm' and '425–600 μm') (Sigma-Aldrich, Merck Millipore, Germany) using a vortex mixer and performing 5 cycles of freezing and thawing with a –80°C freezer. RNA extraction with TRIZOL reagent and a high salt solution for precipitation (plant) (Takara Bio, Japan) was conducted according to the manufacturer's instructions. The quality and quantity of the RNA was verified using an Agilent RNA 6000 Nano Kit on an Agilent Bioanalyzer (Agilent Technologies, California, USA) and a Nanodrop spectrophotometer (Thermo Fisher Scientific, Massachusetts, USA), respectively. Total RNA samples were subjected to library preparation using an NEB Next Ultra RNA Library Prep Kit (New England Biolabs, Ipswich, MA, USA) according to the manufacturer's protocol (NEB #E7530). These mRNA libraries were sequenced in an Illumina NovaSeq6000 (S2 flow cell) in dual flow cell mode with 150-mer paired-end sequences (Filgen Inc, Japan). The raw read data were submitted to DDBJ/EMBL-EBI/GenBank under the BioProject accession number PRJDB12295.

## Transcriptome analysis

A total of 12 libraries were obtained, trimmed, and filtered using the trimmomatic option (ILLUMINA-CLIP:TruSeq3-PE.fa:2:30:10 LEADING:5 TRAILING:5 SLIDINGWINDOW:4:5 MINLEN:25) of the Trinity program. Paired output reads were used for analysis, and de novo assembly was performed using the Trinity program (*Grabherr et al., 2011*) to obtain the transcript sequences. The reads from each library were mapped onto the de novo assembly sequences and read count data, and the transcripts

per million (TPM) were calculated using RSEM (*Li and Dewey, 2011*) with bowtie2 (*Langmead and Salzberg, 2012*).

In this study, RNA-seq produced 550,711,174 reads from the 12 samples (four independent culture flasks under three conditions), yielding 239,047 contigs by de novo assembly using Trinity. The total number of mapped reads of the quadruplicates in the de novo assembled transcriptome dataset were 77,160,394 reads for samples taken before medium change (labelled as 'control'), 74,594,446 for the normal pH culture condition (labelled as 'normal'), and 73,882,605 for the low pH culture condition (labelled as 'lowpH'). Overall, we obtained the count values of the genes in the transcriptome dataset under the three conditions.

Differential gene expression analysis was conducted using the count data as inputs for the R package TCC (*Sun et al., 2013*) to compare the tag count data with robust normalization strategies, with an option using edgeR (*Robinson et al., 2010*) to detect differential expressions implemented in TCC. To identify the DEGs, a false discovery rate (FDR, or q-value) of 0.01, was used as the cutoff.

To annotate the de novo transcript sequences, BLASTp search was performed (E-value cutoff, $10^{-4}$) against the GenBank *nr* database using all the transcript sequences as queries, resulting in 51,833 orthologs. Gene ontology (GO) term annotation of the de novo transcript sequences was performed using InterProScan (*Jones et al., 2014*) ver 5.42–78.0, resulting in 7,336 genes with GO terms. GO term enrichment analysis was performed using the GOseq (*Young et al., 2010*) package in R. Over-represented p-values produced by GOseq were adjusted using the Benjamini-Hochberg correction (*Benjamini and Hochberg, 1995*). An adjusted p-value (q-value) of 0.05, was used to define enriched GO terms. In the KEGG pathway analysis, the ortholog protein sequence obtained via BLASTp search of the DEGs was used as a query. Additionally, KOID was added by blastKOALA (*Kanehisa et al., 2016*) (https://www.kegg.jp/blastkoala/) and mapped to the KEGG pathway using KEGmappar (*Kanehisa and Sato, 2020*) (https://www.genome.jp/kegg/mapper.html). For 'CAZymes in a genome' (CAZome) analysis, all isoform sequences of the DEGs were analysed using CAZy (*Lombard et al., 2014*) (http://www.cazy.org/). For visualization purpose, two outliers TRINITY_DN38357_c4_g1_i9 and TRINITY_DN40801_c4_g1_i5 showing very low expression are not presented in *Figure 3C*.

To compare our results with a previous study using free-living and symbiotic algae (*Xiang et al., 2020*), data were downloaded from NCBI (https://trace.ncbi.nlm.nih.gov; accessions are SRR10578483 and SRR10578484) and analysed in the same way as described earlier. Briefly, expression levels were calculated by RSEM using the de novo assembled references generated in this study, and differential expressed genes were identified by TCC with FDR of 0.01 as the cutoff.

## Cellulase inhibition experiment in vitro

After pre-incubated in normal IMK media for more than one week, *Breviolum* cells were incubated in fresh normal media for 1 day at densities of $4.3 \times 10^7$ cells/20 ml per T25 culture flask with a filter cap (TrueLine Cell Culture Flasks, TR6000). The cells were collected by centrifuging 8 ml of culture medium and ground with two sizes of glass beads (5 and 30 µL volume of '≤106 µm' and '425–600 µm', respectively) (Sigma-Aldrich, Merck Millipore, Germany) in 200 µl Reaction buffer (Cellulase Activity Assay kit, Abcam, UK) using a vortex for 5 min. The homogenates were centrifuged (10,000 *g* at 4°C for 10 min) to collect the supernatants. The supernatants were diluted five times with Reaction buffer and used for measuring cellulase activity. PSG (Biosynth Ltd., United Kingdom) was added to reach a final concentration of 10 and 1 mM. Cellulase activity was conducted according to the manufacturer's instruction (Cellulase Activity Assay kit, Abcam, the UK), using a microplate reader (SH-9000Lab, Hitachi High-Tech Co., Japan) for measurement.

## Cellulase inhibition experiment in vivo

After pre-incubated in normal IMK media for more than one week, *Breviolum* cells were inoculated to fresh low pH media containing 0, 0.1, and 1 mM PSG (Biosynth Ltd., United Kingdom) for 1 day to examine the effect of cellulase inhibitor in monosaccharide secretion. The cells were incubated at densities of $4 \times 10^6$ cells/ml in a 24well plates (n=4 biological replicates for each condition). The supernatant from each culture was collected following centrifugation at 2000×*g* for 2 min at room temperature and filtered using a 0.22 µm PVDF filter (Merck Millipore, Germany).

Glucose and galactose were quantified using a LC–MS/MS system in which a Shimadzu UPLC system (Shimadzu, Kyoto, Japan) was interfaced to an AB Sciex qTrap 5500 mass spectrometer equipped with

an electrospray ionization source (AB SCIEX, Foster, CA, USA). A UK-Amino column (3 µm, 2.1 mm × 250 mm, Imtakt Corporation, Kyoto, Japan) was applied for analysis. Mobile phase A is 0.1% formic acid, and mobile phase B is acetonitrile. Samples (1 µl) were injected and analyzed over a gradient of: 0–0.5 min 95% buffer B (isocratic); 0.5–10 min 85% buffer B (linearly decreasing); 10–15 min 40% buffer B (linear decreasing). The column was equilibrated for 5 min before each sample injection. The flow rate was 0.3 ml/min. Under these analytical conditions, the retention times for glucose and galactose were 11.6 and 10.9 minutes, respectively. Mass spectrometric analysis employed electrospray ionization in the negative mode with multiple reaction monitoring (MRM) at the transitions of m/z 179→89 for glucose and galactose. The optimized MS parameters were as follows: ion spray voltage (–4500 V), declustering potential (–90 V), entrance potential (–10 V), collision energy (–12 V), collision exit potential (–11 V), collision gas ($N_2$ gas) and nebulizer temperature (450°C). Raw data was analyzed using MultiQuant software (AB SCIEX, Foster, CA, USA). Concentrations of monosaccharides were calculated by comparing the peak ratios between the targets of interest and standards.

## Acknowledgements

We thank Dr. Naoki Shinohara and Prof. Fukumatsu Iwahashi for their support in ion chromatography and mass spectrometry, Ms. Yuna Uchida for her assistance in maintaining the algal cultures, Dr. Sara E Milward for her critical reading of the manuscript, Prof. Hiromu Tanimoto and Dr. Vladimiros Thoma for their help in preparing the manuscript, Drs. Kota Kera and Seiji Kojima for their contributions in the initial phase of the project, as well as Profs. John R Pringle and Arthur R Grossman for their generous gift of algal cultures. This work was supported by JSPS KAKENHI (Grant Numbers JP20J01658 [to YI], JP21H05040 [to JM], JP17K15163, JP19H04713, JP19K06786, JP22H05668 and JP22H02697 [to SM]), NIBB Collaborative Research Program 18-321 and 19-332 (to SM), Institute for Fermentation, Osaka (to SM), Program for Creation of Interdisciplinary Research, Frontier Research Institute for Interdisciplinary Sciences, Tohoku University (to SM), and the Gordon & Betty Moore Foundation's Marine Microbiology Initiative #4985 (to JM). Computational resources were provided by the Data Integration and Analysis Facility at the National Institute for Basic Biology and the NIG supercomputer at the ROIS National Institute of Genetics.

## Additional information

### Funding

| Funder | Grant reference number | Author |
| --- | --- | --- |
| Japan Society for the Promotion of Science | JP20J01658 | Yuu Ishii |
| Japan Society for the Promotion of Science | JP21H05040 | Jun Minagawa |
| Japan Society for the Promotion of Science | JP17K15163 | Shinichiro Maruyama |
| Japan Society for the Promotion of Science | JP19H04713 | Shinichiro Maruyama |
| Japan Society for the Promotion of Science | JP19K06786 | Shinichiro Maruyama |
| Japan Society for the Promotion of Science | JP22H05668 | Shinichiro Maruyama |
| Japan Society for the Promotion of Science | JP22H02697 | Shinichiro Maruyama |
| National Institute for Basic Biology | Collaborative Research Program 18-321 | Shinichiro Maruyama |
| National Institute for Basic Biology | Collaborative Research Program 19-332 | Shinichiro Maruyama |

| Funder | Grant reference number | Author |
|---|---|---|
| Institute for Fermentation, Osaka | General Research Grant | Shinichiro Maruyama |
| Frontier Research Institute for Interdisciplinary Sciences, Tohoku University | Program for Creation of Interdisciplinary Research | Shinichiro Maruyama |
| Gordon and Betty Moore Foundation | Marine Microbiology Initiative #4985 | Jun Minagawa |

The funders had no role in study design, data collection and interpretation, or the decision to submit the work for publication.

## Author contributions

Yuu Ishii, Data curation, Formal analysis, Funding acquisition, Validation, Investigation, Visualization, Methodology, Writing – original draft, Project administration, Writing – review and editing; Hironori Ishii, Takeshi Kuroha, Data curation, Formal analysis, Investigation, Methodology; Ryusuke Yokoyama, Ryusaku Deguchi, Kazuhiko Nishitani, Masakado Kawata, Supervision, Validation, Methodology; Jun Minagawa, Supervision, Funding acquisition, Validation, Methodology; Shunichi Takahashi, Data curation, Investigation, Methodology, Writing – original draft, Writing – review and editing; Shinichiro Maruyama, Conceptualization, Resources, Data curation, Software, Formal analysis, Supervision, Funding acquisition, Validation, Investigation, Visualization, Methodology, Writing – original draft, Project administration, Writing – review and editing

## Author ORCIDs

Yuu Ishii http://orcid.org/0000-0003-1735-9557
Takeshi Kuroha http://orcid.org/0000-0002-8327-962X
Ryusuke Yokoyama http://orcid.org/0000-0003-0326-0433
Ryusaku Deguchi http://orcid.org/0000-0003-4571-9329
Kazuhiko Nishitani http://orcid.org/0000-0002-0073-9564
Jun Minagawa http://orcid.org/0000-0002-3028-3203
Masakado Kawata http://orcid.org/0000-0001-8716-5438
Shinichiro Maruyama http://orcid.org/0000-0002-1128-5916

## Decision letter and Author response

Decision letter https://doi.org/10.7554/eLife.80628.sa1
Author response https://doi.org/10.7554/eLife.80628.sa2

# Additional files

## Supplementary files
• MDAR checklist

## Data availability

The raw read data were submitted to DDBJ/EMBL-EBI/GenBank under the BioProject accession number PRJDB12295. To compare our results with a previous study using free-living and symbiotic algae (*Xiang et al., 2020*), data were downloaded from NCBI (https://trace.ncbi.nlm.nih.gov; accessions are SRR10578483 and SRR10578484) .

The following dataset was generated:

| Author(s) | Year | Dataset title | Dataset URL | Database and Identifier |
|---|---|---|---|---|
| Ishii Y, Maruyama S | 2023 | Environmental pH signals the release of monosaccharides from cell wall in coral symbiotic alga | https://ddbj.nig.ac.jp/resource/bioproject/PRJDB12295 | DDBJ Sequence Read Archive (DRA), PRJDB12295 |

The following previously published datasets were used:

| Author(s) | Year | Dataset title | Dataset URL | Database and Identifier |
|---|---|---|---|---|
| Carnegie Institution for Science | 2019 | RNA-Seq of Breviolum minutum SSB01: free-living | https://www.ncbi.nlm.nih.gov/sra/SRR10578483 | NCBI Sequence Read Archive, SRR10578483 |
| Carnegie Institution for Science | 2019 | RNA-Seq of Breviolum minutum SSB01: symbiotic | https://www.ncbi.nlm.nih.gov/sra/?term=SRR10578484 | NCBI Sequence Read Archive, SRR10578484 |

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
