## [Editor Report]

The manuscript makes a fundamental contribution to our understanding of sugar release by symbiotic dinoflagellates and is of broad interest to the fields of ecology, marine biology, and cell biology. The experiments, which combine algal culture with targeted metabolomics, transcriptomics, and the application of inhibitors, provide convincing evidence for an acidic environment mimicking conditions reported for the intracellular organelle that hosts the symbiotic algae, leading to upregulation of algal cellulases, which in turn degrade the algal cell wall and thereby releasing glucose and galactose that can be used as a source of food by the coral host. This is a new idea and could significantly contribute to our understanding of photosymbiosis.

---

## [Decision Letter]

**Decision letter after peer review:**

Thank you for submitting your article "Environmental pH signals the release of monosaccharides from cell wall in coral symbiotic alga" for consideration by *eLife*. Your article has been reviewed by 2 peer reviewers, and the evaluation has been overseen by a Reviewing Editor and Meredith Schuman as the Senior Editor. The reviewers have opted to remain anonymous.

The reviewers and editors agree that your suggested mechanism of sugar secretion of coral symbionts, based on the upregulation and action of cellulases at low pH, is highly interesting and novel. However, we also agree that several important experiments are missing to sufficiently support this hypothesis. Below, we summarize three essential points which must be addressed in your revision. Furthermore, although not essential, please also consider the additional points raised by each of the two reviewers, as these are generally constructive and will likely help to improve your manuscript.

Essential revisions:

1. There is a substantial time difference between the measurements of glucose/galactose and transcriptomics (one day) versus cell wall morphology (three weeks). The prediction would be that the low pH results in increased carbohydrate release via changes in the cell wall morphology, and that the transcript regulation is connected to this. However, this connection is missing and should be better addressed. One possible approach could be to conduct a targeted analysis of gene expression with e.g. qPCR, and measurements of glucose/galactose over time, at steady-state conditions of the acidic environment for a time frame of two-three weeks.

This would also address the concern that any change in carbohydrate release and transcript changes is an acute "shock" response and not specifically connected to the low pH conditions.

2. Another critical experiment addresses the action of the cellulase at low pH. The data show that in algae cultures at low pH, more monosaccharides are present in the media; and that at normal pH plus PSG cellulase inhibitor, fewer monosaccharides are present in the medium. However, the final link is missing. One way to address this would be measuring monosaccharide production at low pH before and after the addition of the PSG inhibitor at varying concentrations.

3. Inhibitor specificity: Chemical interference with biological processes is always very useful and much acknowledged. However, it is also always connected with the concern of unwanted/undetected side effects.

Please address this concern more rigorously (either by additional arguments or experimental data) for the putative cellulase inhibitor PSG, considering questions such as:

What is the evidence that PSG inhibits cellulases, and more specifically, dinoflagellate cellulases (in particular GH7 cellulase)? Is the IC50 known? Is it known whether PSG can affect other enzymes at the concentrations tested (0.1 and 1 mM)? What can be said about the specific mode of action of PSG? Could it also affect other enzymes, like the CAZymes in Figure 3C? The latter question could be addressed by comparing the site and mode of action of the drug across the amino acid sequences of the potential cellulase/CAZyme targets.

*Reviewer #1 (Recommendations for the authors):*

Introduction

– The Introduction is very short and lacks information that is essential for interpreting the study. For example, what is the role of cellulases and their relation to the cell wall? What is known about cellulases in dinoflagellates? As far as know, cellulose is a polymer of β-glucose, so why was galactose predicted to be present in the culture medium, and affected by the PSG treatment?

– A few papers have reported the effect of pH on algal growth in vitro, and should be discussed (e.g. Smith and Muscatine (1999) Marine Biology 134:405-418).

Methods/Results

– More details are needed regarding drug additions and preparations. And some of the info is present throughout the text or figures, but they should be consolidated in the Methods for clarity. For example:

– DCMU was dissolved in ethanol, not DMSO. Potentially, ethanol could have toxic effects or be metabolized and incorporated into the TCA cycle. Was the control treatment also supplemented with ethanol at an equal concentration to that of the DCMU treatment?

– Figure 1-supp1 reports that glucose/galactose in the media was measured after 1 day after DCMU addition. This info should also be reported in Methods.

– Line 81: Burriesci 2012 saw that the transport of newly fixed glucose to the host was stopped with DCMU addition. Additionally, they found specific differences in the secretion of glucose vs. glycerol between free-living algae and those undergoing symbiosis. These are important differences that should be described in more detail in relation to the current study (which only used cultured free-living algae).

– Figure 2: define P, CP, NP

– Figure 2C: Since these are free-living dinoflagellates, they should have a flagellum; however, it is not visible in these images. This raises concerns about the effectiveness of the fixative, or whether these cells really reflect free-living algae.

– Figure 2E/F: these results should be discussed in more detail in the main text. In the figure, further annotation should be provided to illustrate the features of importance of the NP (non-peeled), P (peeled), or CP (completely peeled) morphologies. Is there any relationship between the SEM and TEM observations? (i.e. NP-P-CP vs smooth-wrinkled).

– The Methods state that "73 and 129 cells pooled form n = 4 technical replicates". This would mean that N=1, which would not be acceptable. Please clarify.

– Please show more images of the different morphologies, either in the main manuscript or as supplemental material.

– Figure 3: define "CAZymes" in legend.

– 12 carbohydrate-active enzymes are differentially expressed after 1 day in low pH media yet only cellulase is examined or elaborated upon. Discussion of the others is warranted. Is this pattern maintained after 3 weeks?

– Please provide a list of the other genes that are differentially regulated, and compare it to published transcriptomic analyses or other data from free-living and symbiotic algae.

Discussion

– The major assumption of this study is that the incubations at low pH mimic Breviolum inside the coral symbiosome during symbiosis. However, the majority of the Low pH incubations lasted for only 1 day, and only the morphology observations occurred after 3 weeks.

– For example, Line 128: claims that "this pathway occurred over days". However, most of the experiments involved a 1-day incubation in Low pH, whereas Burriesci et al. 2012 were done using isotopes in established Aiptasia symbiosis.

– Line 133-135: this statement is not supported by the growth data shown in Figure 1A which is slowed under low pH. Maybe the authors mean "inhibit cell proliferation"? In this case, this claim should be tested by inhibiting cellulase in Low pH media and measuring growth rate. It would also be interesting to see the effect of cellulase inhibitors on cell morphology.

– Line 144-146: in my opinion (which matches that of many experts) is that the symbiosome derives from lysosomes and/or phagosomes, which are acidic. It is true that their acidic nature seems to have important implications for symbiosis (e.g. Tang. Front. Microbiol. 6, 816 (2015), but this is very different from "acidifying symbiosomes" being an evolutionary response for algal symbiosis).

Other/General

– What are the dissolved inorganic carbon (DIC) levels in each condition? This is important because pH was adjusted with HCl, which would result in the formation of CO2 that would bubble off resulting in decreased DIC availability for photosynthesis.

– Check all figure legends and axes titles for typos, missing information, and undefined abbreviations.

– I presume the culture media contains vitamins, nitrate, and other compounds that promote algal growth but do not match the conditions found in seawater. This means that the Control cells do not truly reflect free-living algae, which has important consequences for the interpretation of the results and their significance.

*Reviewer #2 (Recommendations for the authors):*

(1) The data on CAZymes and their connection to the narrative of cellulose breakdown and monosaccharide release is very interesting, as the transcriptomics experiment shows the significant upregulation upon low pH of cellulase with a signal peptide (Figure 3). Also intriguing is that the various genes and isoforms display markedly different levels of gene expression changes, both in extent and direction. However, more information about the cellulases is needed for the reader to place these results in context to understand the biology of these molecules, particularly the action and specificity of these enzymes. First, no explicit connection is made between the Trinity ID in the main text and the cellulase gene and isoform labels shown in Figure 3C, although presumably based on context, GH7+CBM1 is the TRINITY_DN40554_c2_g2, so this can be improved by better labeling. Second, the interpretation would benefit from the brief inclusion of information about what distinguishes the various isoforms of these genes, including whether the active sites are conserved or divergent. Finally, related to the CAZymes but on another aspect: are these genes and cellulases found in any other studies of steady-state symbiosis? If this is a common pathway in coral-algal symbiosis, it would likely be present in at least some of the many publicly available datasets of symbiotic algae in hospite, even specifically Breviolum minutum. Therefore, it would be helpful to the reader and to the broader field to place these results in the broader context by outlining connections to the existing literature on these symbiotic systems.

(2) The Discussion mentions several times a connection to cell cycle progression (Line 122). The text states that "photosynthesis-limiting conditions trigger an environmental response in the algae to sustain cell proliferation, through cell cycle progression by cellulase, accompanying cell wall degradation and monosaccharide secretion" (Lines 133-135), as well as a mechanism of "secretion of monosaccharides … to compensate for retarded cell cycle progression" (Line 142-144). However, the 24 hr timepoint is substantially less than the doubling time of the algae, in particular in the low pH medium (Figure 1A), so it would seem there has been little cell division when these cellulases are mobilized at that time point. This claim should be better connected to and substantiated by the data presented.

(3) The cell aggregation results are an interesting phenotype observed, and lead the reader to ask: how does the decreased cell clumping seen in low pH (Figure 2A-B) fit with the concurrent higher monosaccharide secretion (Figure 1), when other bodies of research have suggested that increased monosaccharide secretion contributes to increased cell aggregation (Lines 141-142)?

(4) The data presented in Figure 1 show sugars "secreted", which is described in the Methods as being calculated from the differences in concentration in the media between 24hr vs. 0hr after exposure (Line 204-205). In contrast, the data presented in Figure 4 show the sugar "concentration" in the medium, without the calculation. These varying presentations of the same type of data can be initially confusing, especially when considering how different the baseline sugar concentrations are between the experiments (Figure 4 '0 mM PSG' condition vs. Figure 1 'normal' condition). This could be clarified by showing the raw data of monosaccharide concentrations in all cases, to allow for ease of interpretation, particularly across experiments. Further help to the reader would be to include in the legends that the data in Figure 1 is from ion chromatography and the data from Figure 4 is from LC-MS/MS, which may account for some differences in baseline sugar concentration across these experiments.

(5) While the experiments are excellently described in thorough detail to allow repetition, to fully understand the experimental design required flipping between the Results, Figure Legends, and various sections of the Methods to get a detailed overview of exactly how the experiments were conducted. Thus, the manuscript would greatly benefit from a schematic of the experimental design, perhaps as a small panel in Figure 1. It would be very helpful to show that the algae were grown in rich media, passaged to minimal media IMK for 1 week, then put in the two different pH media and collected after 24 hr (or 3 weeks).

(6) Line 81 and Figure 1: please move the Supp. Figure S1 to be a panel in the main figure. It shows an exciting and important main aspect of the work, and the baseline 'normal' data is the same between figures, as the authors point out.

(7) Lines 158-159: while a knockdown of cellulase would indeed be powerful, the PSG inhibitor here presumably accomplishes the same.

(8) Figure 2: include in the figure or legend the definitions of NP, P, and CP.

(9) Figure 3C:

– The use of asterisks to indicate isoforms is confusing, as this symbol is most commonly used to indicate statistical significance. Use bolding or boxing or some other clear way to highlight these two genes/isoforms as being of particular interest.

– Legend: state here clearly and succinctly the definitions of control, normal, and low pH, as well as the time point of collection.

– Add a title across the graph, e.g. "expression of CAZy isoforms in low pH".

– The expression is by logFC, not log2FC? The color scale, therefore, masks quite a large range of expression differences, especially the potential differences in upregulation of the key cellulase isoforms (all are orange, which according to the key is logFC from 0.0 to 2.5, a large range). Also, the colors of the data points don't match the scale in tone, requiring extra work to match these up. Readability would be greatly improved if the scale was changed to log2FC with a more fine-scale gradation of expression levels, to better highlight the interesting differences in gene expression.

– According to the Methods, the DEGs in the transcriptome analysis were chosen by FDR cutoff of 0.01, but several of these CAZymes fall above that threshold yet are still referred to as DEGs. Further clarification would help the reader.

---

## [Author Response]

Essential revisions:1. There is a substantial time difference between the measurements of glucose/galactose and transcriptomics (one day) versus cell wall morphology (three weeks). The prediction would be that the low pH results in increased carbohydrate release via changes in the cell wall morphology, and that the transcript regulation is connected to this. However, this connection is missing and should be better addressed. One possible approach could be to conduct a targeted analysis of gene expression with e.g. qPCR, and measurements of glucose/galactose over time, at steady-state conditions of the acidic environment for a time frame of two-three weeks.This would also address the concern that any change in carbohydrate release and transcript changes is an acute "shock" response and not specifically connected to the low pH conditions.

We appreciate your critical comments. We apologize that the Materials and methods in our previous manuscript was not comprehensible enough and have revised the section to clarify the followings. First, cell wall morphology analyses by SEM and TEM were done at day 1, not three weeks, after the medium change. Only exceptionally, cell growth and cell clumping examinations took three weeks. Please see Figure 1—figure supplement 1 for schematic representation of the methods. Second, acute “shock” responses on the glucose and galactose secretion were observed at day 0 in the sample “Low pH” or “Normal +DCMU”, presumably due to the shock by medium change procedures or components in new media. However, the differences in the amounts of monosaccharides by these “acute” shocks were not substantial (Figure 1—figure supplement 2). Third, we added new data (Figure 4—figure supplement 1) to examine the effect of the cellulase inhibitor PSG on cellulase enzyme activity, which connects low pH signals, cellulase gene expression, enzyme activity, and monosaccharide secretion. Collectively, we believe that the revised version of the manuscript was better constructed and demonstrated more clearly the connections between carbohydrate release and transcript changes.

2. Another critical experiment addresses the action of the cellulase at low pH. The data show that in algae cultures at low pH, more monosaccharides are present in the media; and that at normal pH plus PSG cellulase inhibitor, fewer monosaccharides are present in the medium. However, the final link is missing. One way to address this would be measuring monosaccharide production at low pH before and after the addition of the PSG inhibitor at varying concentrations.

We sincerely apologize the confusion that our previous manuscript may have made. In the previous and current manuscripts, we measured monosaccharide release at low pH, not normal, with the addition of the inhibitor PSG at varying concentrations, as you suggested. To make it clearer, we revised the whole Materials and methods section, which now shows the link between pH effect, PSG action, and cellulase activity.

3. Inhibitor specificity: Chemical interference with biological processes is always very useful and much acknowledged. However, it is also always connected with the concern of unwanted/undetected side effects.Please address this concern more rigorously (either by additional arguments or experimental data) for the putative cellulase inhibitor PSG, considering questions such as:What is the evidence that PSG inhibits cellulases, and more specifically, dinoflagellate cellulases (in particular GH7 cellulase)? Is the IC50 known? Is it known whether PSG can affect other enzymes at the concentrations tested (0.1 and 1 mM)? What can be said about the specific mode of action of PSG? Could it also affect other enzymes, like the CAZymes in Figure 3C? The latter question could be addressed by comparing the site and mode of action of the drug across the amino acid sequences of the potential cellulase/CAZyme targets.

To our knowledge, there have been no specific molecular mechanisms of cellulase inhibitors reported so far. PSG is a sugar analog and was used as a potent inhibitor of β-glucosidase in a previous study (Yoshida 1995 Plant Cell Reports) and references therein. Yoshida (1995), now cited in the revised manuscript, showed that 10 mM PSG inhibited cellulases, β-glucosidases, and α-arabinosidases in rice, suggesting that other types of enzymes and cellulases in other species could be targets of PSG in similar concentration ranges. In general, CAZyme activities and targets are difficult to predict by sequence similarity due to high divergence in sequences and have been little biochemically studied so far, thus we believe that the effects of PSG on other enzymes can be very interesting subjects of future studies.

To gain insights into dinoflagellate cellulase activities, we conducted in vitro assays to show that PSG inhibits cellulase activity in the whole cells, not selectively secreted enzymes in Figure 4—figure supplement 1, in the revised manuscript. We used the whole cell partly because secreted cellulase activities were very low and under detection limits. Although this does not necessarily indicate that PSG specifically inhibits dinoflagellate GH7 cellulases which we featured in the manuscript, our data showed that dinoflagellate cellulase activities and the inhibitory effects by PSG could be quantitatively measured.

Reviewer #1 (Recommendations for the authors):Introduction– The Introduction is very short and lacks information that is essential for interpreting the study. For example, what is the role of cellulases and their relation to the cell wall? What is known about cellulases in dinoflagellates? As far as know, cellulose is a polymer of β-glucose, so why was galactose predicted to be present in the culture medium, and affected by the PSG treatment?

We appreciate the helpful comments from Reviewer 1. We revised the Introduction section to state that cellulases are critical to maintain cell cycle progression in dinoflagellates and cited relevant references. We also added information on cellulases in dinoflagellates in the Discussion section, as follows:

Line 148 in the revised manuscript:

“Previous studies showed that cell wall degradation/rearrangement by cellulase is required for cell cycle progression (Kwok and Wong, 2010) and cellulose synthesis is involved in morphogenesis (Chan et al., 2019) in dinoflagellates.”

Regarding galactose, detection of galactose secreted in the media was a big surprise to us. To our knowledge, while it is generally accepted that dinoflagellate cell walls contain cellulose, there have been no studies which clarified the molecular details and the whole diversity of glycan families compositing dinoflagellate cell walls. In addition, cellulose is a polymer of glucose as you suggested, but not the only target of cellulases. Now we directly confirmed that PSG inhibited cellulase enzyme activity in vitro, suggesting that galactose secretion and cellulase activity are at least correlated. Although it is unknown how galactose secretion is increased in low pH, one plausible explanation is that the cell wall contains complex galactose-containing glycans and that cellulase degrades main glucan chain to make galactose residues accessible to other CAZymes. Now we discussed this in the revised manuscript:

Line 170:

“In this study, we show that Breviolum secrets galactose as well as glucose (Figure 1). Although the mechanism of action of galactose secretion is unknown, less substantial increase of galactose secretion under low pH (Figure 1) and the significant inhibitory effect of PSG (Figure 4) suggest that galactose secretion may be regulated by uncharacterized PSG-sensitive enzymes. Under low pH, multiple upregulated genes encoding CAZymes that break down glycosidic bonds (e.g. chitinase, hexosaminidase, mannosidase) were detected (Figure 3). The cell wall components of Symbiodiniaceae are unknown, but complex galactose-containing glycans that constitute the cell wall may be targets of these CAZymes.”

– A few papers have reported the effect of pH on algal growth in vitro, and should be discussed (e.g. Smith and Muscatine (1999) Marine Biology 134:405-418).

We appreciate the comment. Unfortunately, we could not find previous papers specifically reporting the effect of pH, including Smith and Muscatine 1999, as basically they detected or discussed the effect of molecules which were supposed to affect the pH (e.g. nitrogen compounds, CO2, amino acids). Simple ingredients can affect not only pH but also physiology as a whole and the interpretation of the effects is very complex, so that these studies are beyond the scope of this study.

Methods/Results– More details are needed regarding drug additions and preparations. And some of the info is present throughout the text or figures, but they should be consolidated in the Methods for clarity. For example:– DCMU was dissolved in ethanol, not DMSO. Potentially, ethanol could have toxic effects or be metabolized and incorporated into the TCA cycle. Was the control treatment also supplemented with ethanol at an equal concentration to that of the DCMU treatment?

We understand your concerns and thoroughly revised the Methods section. Regarding DCMU, we added the same concentration of ethanol in the control samples, as you pointed out, and described this in the Methods, Line 245.

– Figure 1-supp1 reports that glucose/galactose in the media was measured after 1 day after DCMU addition. This info should also be reported in Methods.

We appended the information in the Methods, Line 248.

– Line 81: Burriesci 2012 saw that the transport of newly fixed glucose to the host was stopped with DCMU addition. Additionally, they found specific differences in the secretion of glucose vs. glycerol between free-living algae and those undergoing symbiosis. These are important differences that should be described in more detail in relation to the current study (which only used cultured free-living algae).

We appended this information, considering that Burriesci 2012 discussed mainly glycerol secretion using cultured free-living algae, as follows:

Line 95:

“On the addition of the photosynthesis inhibitor 3-(3,4-dichlorophenyl)-1,1-dimethylurea (DCMU), the concentrations of glucose and galactose in the medium increased (Figure 1E, F, Figure 1—figure supplement 2), suggesting the presence of a pathway uncharacterized in previous studies, where the transport of newly fixed glucose, not glycerol, to the host sea anemone was blocked by DCMU addition (Burriesci et al., 2012).”

– Figure 2: define P, CP, NP.

We appended this information in the Figure 2 legend.

– Figure 2C: Since these are free-living dinoflagellates, they should have a flagellum; however, it is not visible in these images. This raises concerns about the effectiveness of the fixative, or whether these cells really reflect free-living algae.

This is an important point in this experiment: As shown in a previous study in which the culture strain Breviolum sp. SSB01 was originally isolated, the stain used has been isolated through screenings on agar plates, and most of the cells are maintained as coccoids. This strain was used partly because most of other strains contain contaminated/co-cultured bacteria and were not usable for biochemical and physiological assays in this study. In SEM and TEM analyses, the ratios of flagellated cells were low and most of the micrographs were non-flagellated.

– Figure 2E/F: these results should be discussed in more detail in the main text. In the figure, further annotation should be provided to illustrate the features of importance of the NP (non-peeled), P (peeled), or CP (completely peeled) morphologies. Is there any relationship between the SEM and TEM observations? (i.e. NP-P-CP vs smooth-wrinkled).

We totally agree with your comment; we added annotations of NP, P, CP in Figure 2 legend and discussed the cell surface structures more in detail in the Discussion section, as follows.

Line 150:

“Indeed, wrinkle and exfoliation of the algal cell wall was observed under low pH conditions using SEM and TEM, respectively, suggesting that the cell walls are morphologically and qualitatively modified under low pH in Breviolum (Figure 2).”

In our view, relationships between SEM and TEM results were very important and interesting, but unfortunately, while they apparently show some similarity and correspondence (e.g. more NP and smooth cells in normal condition), it is difficult to conclude that there are some clear relationships, since exactly the same cell could not be observed using SEM and TEM. This should be examined in a future study.

– The Methods state that "73 and 129 cells pooled form n = 4 technical replicates". This would mean that N=1, which would not be acceptable. Please clarify.

We presented biological replicates (N=2) in the revised manuscript, in Figure 2.

– Please show more images of the different morphologies, either in the main manuscript or as supplemental material.

We added more images in Figure 2—figure supplement 1 and 2.

– Figure 3: define "CAZymes" in legend.

“Carbohydrate-Active enZymes” was appended in the legend.

– 12 carbohydrate-active enzymes are differentially expressed after 1 day in low pH media yet only cellulase is examined or elaborated upon. Discussion of the others is warranted. Is this pattern maintained after 3 weeks?

We totally agreed and appended further discussion of other enzymes, although molecular studies are very limited, in the Discussion section Line 171:

“Although the mechanism of action of galactose secretion is unknown, less substantial increase of galactose secretion under low pH (Figure 1) and the significant inhibitory effect of PSG (Figure 4) suggest that galactose secretion may be regulated by uncharacterized PSG-sensitive enzymes. Under low pH, multiple upregulated genes encoding CAZymes that break down glycosidic bonds (e.g. chitinase, hexosaminidase, mannosidase) were detected (Figure 3). The cell wall components of Symbiodiniaceae are unknown, but complex galactose-containing glycans that constitute the cell wall may be targets of these CAZymes.”

Again, we apologize the confusion made from the previous version of the manuscript, “3 weeks” is the duration of pre-culturing before measurements in most cases. In this study we focused on shorter-term responses (e.g. one day), which we believe are suitable for biochemical and physiological experiment settings we used. Longer-term responses are also important and will be subjects of future studies using more stable culturing systems, e.g. chemostat, bioreactor.

– Please provide a list of the other genes that are differentially regulated, and compare it to published transcriptomic analyses or other data from free-living and symbiotic algae.

We appended the results with re-analyzing the data by Xiang et al. (2020 Nature Comm 11: 108) as an additional column in Figure 3-source data 1 “Expression levels of the annotated low pH DEGs”.

Discussion– The major assumption of this study is that the incubations at low pH mimic Breviolum inside the coral symbiosome during symbiosis. However, the majority of the Low pH incubations lasted for only 1 day, and only the morphology observations occurred after 3 weeks.

We apologize the confusion made from the previous version of the manuscript. First, in this study we did not assume that low pH mimics symbiosome, but rather our assumption is that low pH is a ubiquitous environmental signal in nature and different kinds of animal guts and that many unknown, more complex, host-derived signals may be present in symbiosomes. Second, most experiments including morphological observations are done in one day, not 3 weeks, which was a pre-culture duration in most cases, as explained earlier. We clarified this point in the Materials and methods in the revised manuscript.

– For example, Line 128: claims that "this pathway occurred over days". However, most of the experiments involved a 1-day incubation in Low pH, whereas Burriesci et al. 2012 were done using isotopes in established Aiptasia symbiosis.

Thank you for pointing this out. We changed "this pathway occurred over days" to "this pathway occurred in a day". Burriesci et al. 2012 used an established symbiosis system as you mentioned. However, they measured the metabolite transfer in a shorter time range (5 min to 1 hour) and did not report isotope relocations in a day or longer. This does not necessarily mean that the cellulase-mediated pathway does not work in an hour or that Burriesci et al. 2012 shows that photosynthates will not stay in symbiont cells for days. Rather, we just clarified the differences in the detection time ranges in this and previous studies.

Considering the differences of detection time ranges and the responses to DCMU treatments, our data and Burriesci et al. 2012 pointed towards different sites of action.

– Line 133-135: this statement is not supported by the growth data shown in Figure 1A which is slowed under low pH. Maybe the authors mean "inhibit cell proliferation"? In this case, this claim should be tested by inhibiting cellulase in Low pH media and measuring growth rate. It would also be interesting to see the effect of cellulase inhibitors on cell morphology.

We apologize the confusion. Kwok and Wong 2010 Plant Cell showed that addition of exogenous cellulase enzyme led to cell cycle advancement in the dinoflagellate Crypthecodinium cohnii. Based on this report, our original intention was to say that, when low pH inhibits cell cycle progression, the cells secrete cellulases which can potentially play a role in complementing the retarded cell cycle. Notably, in Kwok and Wong 2010, the relationship between cellulase activity and cell cycle progression was clearly demonstrated but how these are relevant to the cell growth rate (both in terms of cell ‘size’ and ‘number’ growth rates) remains unclear. Considering this, we changed this part to the following:

Line 161:

“This suggests the photosynthesis-limiting conditions trigger an environmental response in the algae to compensate for retarded cell cycle progression by upregulating multiple genes including the one encoding cellulase, accompanying cell wall degradation and monosaccharide secretion (or “leakage”)*.*”

– Line 144-146: in my opinion (which matches that of many experts) is that the symbiosome derives from lysosomes and/or phagosomes, which are acidic. It is true that their acidic nature seems to have important implications for symbiosis (e.g. Tang. Front. Microbiol. 6, 816 (2015), but this is very different from "acidifying symbiosomes" being an evolutionary response for algal symbiosis).

We apologize the confusion and agree with you. We changed this to the following:

Line 180:

“Therefore, acidic symbiosomes may be of evolutionary advantage for cnidarian hosts to promote environmental responses of algal symbionts, which enables monosaccharides to be efficiently secreted within the organelle.”

Other/General– What are the dissolved inorganic carbon (DIC) levels in each condition? This is important because pH was adjusted with HCl, which would result in the formation of CO2 that would bubble off resulting in decreased DIC availability for photosynthesis.

Although we have not measured it in this study due to technical difficulties, lowered photosynthesis speed in low pH (Figure 1) suggests that DIC may be decreased and affect the rate of conversion from DIC to CO2 and photosynthetic carbon fixation. This will be a subject of future studies.

– Check all figure legends and axes titles for typos, missing information, and undefined abbreviations.

Thank you for your suggestions. We have carefully revised the manuscript.

– I presume the culture media contains vitamins, nitrate, and other compounds that promote algal growth but do not match the conditions found in seawater. This means that the Control cells do not truly reflect free-living algae, which has important consequences for the interpretation of the results and their significance.

We agree that culture conditions we used do not truly reflect free living conditions in nature, and believe that this is also true in most of algal cultures used in previous studies. Our aim of this study is not to simulate natural condition in laboratory, but to examine the effect of pH by comparing the cultures in media with different pH values.

Reviewer #2 (Recommendations for the authors):(1) The data on CAZymes and their connection to the narrative of cellulose breakdown and monosaccharide release is very interesting, as the transcriptomics experiment shows the significant upregulation upon low pH of cellulase with a signal peptide (Figure 3). Also intriguing is that the various genes and isoforms display markedly different levels of gene expression changes, both in extent and direction. However, more information about the cellulases is needed for the reader to place these results in context to understand the biology of these molecules, particularly the action and specificity of these enzymes. First, no explicit connection is made between the Trinity ID in the main text and the cellulase gene and isoform labels shown in Figure 3C, although presumably based on context, GH7+CBM1 is the TRINITY_DN40554_c2_g2, so this can be improved by better labeling. Second, the interpretation would benefit from the brief inclusion of information about what distinguishes the various isoforms of these genes, including whether the active sites are conserved or divergent. Finally, related to the CAZymes but on another aspect: are these genes and cellulases found in any other studies of steady-state symbiosis? If this is a common pathway in coral-algal symbiosis, it would likely be present in at least some of the many publicly available datasets of symbiotic algae in hospite, even specifically Breviolum minutum. Therefore, it would be helpful to the reader and to the broader field to place these results in the broader context by outlining connections to the existing literature on these symbiotic systems.

We appreciate your helpful comments and revised the text as follows:

i. Lines 116-123 in the revised manuscript, we improved the labeling and explanations; cellulase as an enzyme, GH7+CBM1 as a sequence motif, and TRINITY_DN40554_c2_g2 as a gene model based on the transcriptome assembly.

ii. We re-analyzed the transcriptome of Xiang et al. (2020 Nature Communications) and found that the cellulase gene was differentially expressed in steady-state symbiosis (Figure 3- source data 1).

We believe that, although cellulases are not well annotated with simple similarity search results and not well discussed through published data, the CAZyme profiling in this study enabled to shed light on a connection between cell walls and environmental responses, and symbiosis.

(2) The Discussion mentions several times a connection to cell cycle progression (Line 122). The text states that "photosynthesis-limiting conditions trigger an environmental response in the algae to sustain cell proliferation, through cell cycle progression by cellulase, accompanying cell wall degradation and monosaccharide secretion" (Lines 133-135), as well as a mechanism of "secretion of monosaccharides … to compensate for retarded cell cycle progression" (Line 142-144). However, the 24 hr timepoint is substantially less than the doubling time of the algae, in particular in the low pH medium (Figure 1A), so it would seem there has been little cell division when these cellulases are mobilized at that time point. This claim should be better connected to and substantiated by the data presented.

This is a great point. Kwok and Wong 2010 Plant Cell showed that addition of exogenous cellulase enzyme led to cell cycle advancement in the dinoflagellate Crypthecodinium cohnii. Notably, in Kwok and Wong 2010, the relationship between cellulase activity and cell cycle progression was clearly demonstrated but how these are relevant to the cell growth rate (both in terms of cell ‘size’ and ‘number’) remains unclear. Thus, it is fair to say that cell wall and cell cycle most likely have some connections but the effect of cell wall degradation to cell division (as a result of cell cycle progression) is not fully understood. To make this clear and tone down the direct effects of cell wall to cell division, we revised the Discussion as follows:

Line 161:

“This suggests the photosynthesis-limiting conditions trigger an environmental response in the algae to compensate for retarded cell cycle progression by upregulating multiple genes including the one encoding cellulase, accompanying cell wall degradation and monosaccharide secretion (or “leakage”)”.

(3) The cell aggregation results are an interesting phenotype observed, and lead the reader to ask: how does the decreased cell clumping seen in low pH (Figure 2A-B) fit with the concurrent higher monosaccharide secretion (Figure 1), when other bodies of research have suggested that increased monosaccharide secretion contributes to increased cell aggregation (Lines 141-142)?

Although there have been literally no studies clarifying molecular details of the relationship between secreted saccharides and cell aggregation in dinoflagellates, we think that molecular species and structures of secreted saccharides may vary depending on multiple factors, e.g. lineage, species, kind and strength of environmental stimuli. In Breviolum, low pH signals increased monosaccharide secretion and decreased cell clumping, while other stressors may bring different phenotypes. To clarify this, we cited Kwok et al. 2023 and revised the Discussion section:

Line 168:

“Some dinoflagellates are also known to secrete viscous substances, including monosaccharides, as an environmental response, likely for cell aggregation and biofilm formation (Kwok et al., 2023; Mandal et al., 2011). In this study, we show that Breviolum secrets galactose as well as glucose (Figure 1). … The cell wall components of Symbiodiniaceae are unknown, but complex galactose-containing glycans that constitute the cell wall may be targets of these CAZymes.”

(4) The data presented in Figure 1 show sugars "secreted", which is described in the Methods as being calculated from the differences in concentration in the media between 24hr vs. 0hr after exposure (Line 204-205). In contrast, the data presented in Figure 4 show the sugar "concentration" in the medium, without the calculation. These varying presentations of the same type of data can be initially confusing, especially when considering how different the baseline sugar concentrations are between the experiments (Figure 4 '0 mM PSG' condition vs. Figure 1 'normal' condition). This could be clarified by showing the raw data of monosaccharide concentrations in all cases, to allow for ease of interpretation, particularly across experiments. Further help to the reader would be to include in the legends that the data in Figure 1 is from ion chromatography and the data from Figure 4 is from LC-MS/MS, which may account for some differences in baseline sugar concentration across these experiments.

As suggested, we used “secretion rate (μg/ml/day)” or “concentration (μg/ml)” throughout the manuscript and included the measuring methods (ion chromatography or LC-MS/MS) in the legends of Figures 1 and 4.

(5) While the experiments are excellently described in thorough detail to allow repetition, to fully understand the experimental design required flipping between the Results, Figure Legends, and various sections of the Methods to get a detailed overview of exactly how the experiments were conducted. Thus the manuscript would greatly benefit from a schematic of the experimental design, perhaps as a small panel in Figure 1. It would be very helpful to show that the algae were grown in rich media, passaged to minimal media IMK for 1 week, then put in the two different pH media and collected after 24 hr (or 3 weeks).

We revised the Materials and methods to make it more straightforward and added a panel showing a schematic of the experimental design in Figure 1-S1.

(6) Line 81 and Figure 1: please move the Supp. Figure S1 to be a panel in the main figure. It shows an exciting and important main aspect of the work, and the baseline 'normal' data is the same between figures, as the authors point out.

As suggested, we move Supp Figure S1 in the previous manuscript to the main Figure 1 in the revised version.

(7) Lines 158-159: while a knockdown of cellulase would indeed be powerful, the PSG inhibitor here presumably accomplishes the same.

We agree that our inhibitor experiments presumably work in a similar way as a knockdown. Considering the recently published paper (Gornik et al., 2022) and advances of genetic engineering techniques, we changed “knockdown” to “knockout” (Line 195), which should bring the research one step further.

(8) Figure 2: include in the figure or legend the definitions of NP, P, and CP.

The definitions of NP (non-peeled), P (peeled) and CP (completely-peeled) were included in the legend as suggested.

(9) Figure 3C:– The use of asterisks to indicate isoforms is confusing, as this symbol is most commonly used to indicate statistical significance. Use bolding or boxing or some other clear way to highlight these two genes/isoforms as being of particular interest.

We removed the asterisks and modified the figure to highlight cellulase isoforms, as suggested.

– Legend: state here clearly and succinctly the definitions of control, normal, and low pH, as well as the time point of collection.

As suggested, we explained the definitions of the sample labels and time point in Figure 3A legend as follows, not 3C, as we see that the 3A used the same labeling and better to be explained earlier.

“A. Venn diagram showing the numbers of DEGs between the cells incubated under normal condition for 1 day vs control (before incubation) and between the ones incubated at low pH vs control.”

– Add a title across the graph, e.g. "expression of CAZy isoforms in low pH".

The title “Expression of CAZyme isoforms” was added in Figure 3c.

– The expression is by logFC, not log2FC? The color scale, therefore, masks quite a large range of expression differences, especially the potential differences in upregulation of the key cellulase isoforms (all are orange, which according to the key is logFC from 0.0 to 2.5, a large range). Also the colors of the data points don't match the scale in tone, requiring extra work to match these up. Readability would be greatly improved if the scale was changed to log2FC with a more fine-scale gradation of expression levels, to better highlight the interesting differences in gene expression.

We agreed with the suggestion and revised the Figure 3c for improving readability. In doing this, A few outliers (showing very low expression, low FC and high FDR) were removed, as described in the Methods section, Line 332.

“For visualization purpose, two outliers TRINITY_DN38357_c4_g1_i9 and TRINITY_DN40801_c4_g1_i5 showing very low expression are not presented in Figure 3C.”

– According to the Methods, the DEGs in the transcriptome analysis were chosen by FDR cutoff of 0.01, but several of these CAZymes fall above that threshold yet are still referred to as DEGs. Further clarification would help the reader.

We thank the reviewer for pointing this out. For clarification we revised the Figure 3c legend, considering the followings: Our transcriptome assembly generated by the Trinity software contains two levels of assembles, ‘gene’ and ‘isoform’, and the gene is basically a cluster of the isoforms, each of which can be translated into distinct polypeptides. Using these assembles, we could obtain differentially expressed genes (DEGs) and isoforms (DEIs), but mainly used the former to discuss the expression regulations. However, we found that some isoforms possessed specific protein domains or signal peptides, but others did not, which is critical in discussing potential intracellular localization of the gene products, i.e. cellulases and other CAZymes. Thus, we identified DEGs which were included in the CAZymes, and took out isoform information for these DEGs including the FDR scores from the expression analysis. This results in that some isoforms are fall below the threshold (DEIs) and others are not.